# A Soil Screening Study to Evaluate Soil Health for Urban Garden Applications in Hartford, CT

Hayley Clos [1,*], Marisa Chrysochoou [1], Nefeli Bompoti [1] and Jacob Isleib [2]

1   Department of Civil and Environmental Engineering, University of Connecticut, Storrs, CT 06268, USA
2   United States Department of Agriculture Natural Resources Conservation Service, Tolland, CT 06084, USA
*   Correspondence: hayley.clos@uconn.edu

**Abstract:** Urban agriculture is a sustainable practice for communities to have access to healthy and affordable produce by reducing the energy costs of food production and distribution. While raised beds are often used in community gardens to ensure that soil quality meets proper standards, the use of existing urban soils is desired for economic and sustainability purposes. The main objective of this study is to evaluate a methodology to test soil health parameters using in situ screening methods. Soil testing was conducted at three urban lots in Hartford, CT, that were candidates for community gardens. In situ measurements of metals were taken with a pXRF instrument in all three lots, and an additional 30 samples were tested in the laboratory, both on pressed pellets via pXRF and with acid digestion and ICP-MS analysis. Ultimately, in situ pXRF measurements were comparable to pelletized pXRF and ICP-MS measurements for elements of interest, and pXRF is shown to be a reliable screening tool to evaluate exceedances for metal regulatory thresholds exceeding 100 ppm (e.g., Pb, Cu, Ni, Zn, and Se), although soil moisture content exceeding 5% is shown to have a dilution effect on in situ results up to about a 30% difference. The current study serves as a case study in Hartford, CT, for the evaluation of in situ pXRF analysis as a rapid soil screening tool, and further research will be needed to extend the current recommendations to a general rapid soil assessment methodology.

**Keywords:** urban agriculture; soil health; in situ screening; X-ray fluorescence; trace metals

## 1. Introduction

Urban agriculture is a sustainable practice for communities to combat inflation and provide employment opportunities while serving as an accessible industry for low-income entrepreneurs, improve food security and resolve hunger and nutrition as a major public health concern, and promote personal wellness through plant–human interaction [1,2]. It is estimated that a 10 × 10 m plot under average growing conditions in a 130-day growing season can fulfill an average household's yearly produce and most nutrition requirements [3]. Further representing the economic benefits of urban agriculture, urban farms are more focused on high-value production and have been shown to produce more per acre than rural farms despite their difference in size [4]. Overall, urban agriculture offers economic, environmental health, and community development benefits, including green-space preservation, air quality improvement, reduction of freshwater consumption, and opportunities for compost use and waste reduction [5–12].

The success of urban agriculture is influenced by the quality of soil available at the site. Soil health and fertility are dictated by a number of parameters that can be analyzed and monitored before and during land use conversion, such as organic carbon and nitrogen content, pH, biological/microbial activity, cation exchange capacity, and total clay [13–15]. Standard agricultural soils consist of a balance of mineral components (sand, silt, and clay), organic matter, air, and water, which is impacted by the climate, topography, soil organisms, and parent material at the site. On the contrary, urban gardens are typically created on

unused or vacant land with an industrial history, or impacted by urban fill, and can contain various organic contaminants, usually tied to common pollution sources such as coal ash and bituminous materials in urban fill, as well as inorganic contaminants such as trace metals [16]. Humans can be exposed to those contaminants via inhalation, ingestion, and absorption or direct contact with soils as well as through the plant uptake pathway and consumption of contaminated crops [17,18]. The USEPA [19] outlines the risk assessment parameters for uptake into homegrown produce using models that represent the pathways of soil-to-root uptake, translocation from roots to aboveground plant parts, and atmospheric deposition of contaminants. Leafy and root vegetables common to residential gardens are assumed to have the most uptake capability of contaminants based upon the current models [19–21]. Overall, a sustainable approach to creating urban gardens is to utilize existing soils instead of remedies such as excavation, soil/ground covers, phytoremediation, raised beds, and the application of chelating agents [20,22–25]. Therefore, it is essential to properly screen urban soils specifically intended for agricultural use, which are a concern to human health, for harmful inorganic and organic contaminants.

Currently, soil screening of trace metals occurs via field application of portable X-ray fluorescence (pXRF) spectroscopy, or laboratory analysis utilizing XRF or acid digestion coupled with inductively coupled plasma mass spectrometry (AD/ICP-MS). XRF is often utilized as a greener, faster, more cost-effective, and non-destructive method of analysis compared to AD/ICP-MS. However, the accuracy of in situ pXRF analysis can be impacted by soil matrix interferences such as moisture content leading to dilution, sample heterogeneity, and particle size effects [26]. While the literature does utilize AD/ICP-MS analysis to confirm XRF measurements of soils, very little research has been carried out to determine the effect of different XRF sample preparation methods on the accuracy of trace metal measurements in soils. Parsons et al. [27] compared three pXRF preparation methods to AD/ICP-MS measurements of arsenic in soils from a floodplain in France, though true in situ pXRF analyses were not conducted. The different pXRF preparation methods consisted of in situ field preparation where soils were first homogenized, sieved to <2 mm, and compacted; ex-situ LDPE bags where homogenized and sieved soil material was analyzed in 50 μm thick LDPE bags; and finally, ex situ XRF cups where soil material was homogenized, sieved, dried, ground, and analyzed through Mylar film. While it was determined that all versions of pXRF arsenic measurements are comparable to ICP-MS measurements ($R^2 > 0.85$), pXRF analysis was not applied on raw, unprepared soil samples [27]. Furthermore, there have been studies that analyze the effect of pellet preparation methods for XRF analysis of soils by comparing results from pressed pellets, pressed pellets with binder, and loose powdered samples [28,29]. Finally, in a study by Weindorf et al. [30], in situ pXRF analysis of surface soils, which normally have lower trace element concentrations than subsoils, was compared to AD/ICP atomic emission spectroscopy (AES) analysis, where corrections for original moisture content were made to ICP measurements to represent field conditions under which the pXRF scans were performed. Ultimately, it was determined that corrected ICP measurements matched reasonably well to pXRF measurements for the majority of the studied trace elements (As, Co, Cu, Fe, Mn, Pb, and Zn) excluding Ba and Cr [30].

Currently, there is no guidance in the United States and Europe that unifies such findings to recommend a set of best practices for in situ screening of urban soils for conversion to gardens. The United States Department of Agriculture Natural Resource Conservation Service (USDA-NRCS) often receives requests to provide screening and evaluate the soil health; with over 2300 offices nationwide, the need to provide such services at low cost and turn-around time along with guidance to local farmers is acute. Accordingly, the purpose of this study was to determine if in situ XRF spectroscopy could be applied for accurate determination of trace metal content in urban soils by comparing results to laboratory-prepared pellets using XRF and the industry standard of AD/ICP-MS. In order to further determine the state of soil fertility, ex situ ECS 4020 CHNSO isotope analysis was used to quantify the total carbon and nitrogen content of urban soils.

## 2. Materials and Methods

### 2.1. Site Locations and Sampling Methodology

The site selected for testing was 138 Irving Street in Hartford, CT (41°46′49.8″ N 72°41′09.0″ W), which is approximately 639 m² in size and is located on a street corner next to a residential building. The lot contains no existing structures, and based on the soils present, it was determined that sampling would occur at 0–15 cm and 15–30 cm depths. The lot was divided into a grid of 3 m intervals before visiting the location using the geographic information system (GIS) software version 3.0.3. In the field, a tape measure was used to measure the length of the lot and flags were placed in 3 m intervals along the tape to indicate where sampling would occur. There was a total of 65 testing spots within the site, which were analyzed with in situ pXRF and then collected for laboratory analyses. At each spot, a hand auger and/or shovel was used to obtain soil samples at two depths of 0–15 cm and 15–30 cm, which were measured with a ruler to ensure accuracy. The two soil samples were placed on top of individually labeled plastic sample bags for in situ analyses, and then placed in bags to be returned to the lab for further analysis. The soil measured in situ in this lot was observed to have relatively low moisture, with minimal effect on the obtained readings. Additional soil samples were obtained from two other lots, 176 Clark Street (41°47′23.0″ N 72°40′35.5″ W) and 438 Garden Street (41°46′53.3″ N 72°41′04.2″ W) in Hartford, CT, in another sampling event that took place after a significant rain event. The effect of moisture content on in situ pXRF readings was tested using 32 collective soil samples from the two sites tested during this event. After conducting in situ pXRF analysis using the same method as in the first site, samples were immediately bagged and brought back to the laboratory for a moisture content analysis.

### 2.2. Sample Preparation

Of the 130 soil samples collected from the first site, 15 sampling locations, or 30 total samples, were selected for further analysis based on the comparatively higher elemental concentrations of Pb. The samples were stored in sealed plastic sampling bags at room temperature. A subsample of 30–50 g was then placed into petri dishes, allowed to dry under a fume hood for 48 hours, and passed through a 2 mm sieve. Approximately 10 grams of each dried and sieved sample were pulverized for 5 minutes using a SPEX SamplePrep ShatterBox 8530 (Metuchen, NJ, USA) and sieved to a size of 0.075 mm. A polymer-based binder was added to the pulverized and sieved soil in a 9:1 ratio, homogenized with a LabRAM Resodyn mixer, and pressed into a pellet with a SPEX 3636 X-Press (Metuchen, NJ, USA). All 30 pellets were then analyzed once with the pXRF instrument using the same method as in the in situ scenario. The dried and sieved material of the 30 samples was also sent out to Complete Environmental Testing in Stamford, CT for AD/ICP-MS analysis of selected metals (Pb, Cr, Se, and As).

To evaluate the effect of moisture content in the wet samples from the second and third lot, the subsamples that were air-dried for 48 hours were re-weighed to obtain a value for moisture content, placed into a petri dish, and covered with Mylar film to be analyzed via pXRF in triplicate, simulating in situ analysis conditions for dry soil.

### 2.3. Analytical Methods

Prior to in situ analysis of excavated soils in all three lots in the study, Mylar film was placed over the soil samples and an Olympus Vanta VMR pXRF instrument was used to measure the in situ-based elemental concentrations of each soil sample. The Olympus Vanta VMR pXRF is equipped with a 4-watt rhodium X-ray tube and beryllium window, with a 3 mm diameter beam spot. Three spots of the sample were tested with a one-minute running time each using the instrument's Geochem calibration method, after performing a successful calibration check using the XRF metal standard stainless steel grade 316. The Geochem method was also used to analyze the pelletized samples from the first lot with a single 1 min scan.

Apart from trace-metal analysis, total carbon and nitrogen content of the same 30 samples that were used for pXRF pellets and AD/ICP-MS analysis was determined using the Costech ECS 4010 CHNSO analyzer (Costech Analytical Technologies, INC, Valencia, CA, USA). Approximately 35–40 mg of the pure pulverized soil material was rolled into a tin capsule to be analyzed with the instrument in duplicate by ignition. Analysis also included atropine standards ranging from 0.5–2.5 mg and atropine quality control checks every 12 samples.

### 2.4. Statistical Analyses

Statistical analyses using fitted models and hypothesis testing were performed to compare AD/ICP-MS and the two sets of pXRF measurements. XRF values below the detection limit were replaced by the margin of error σ provided by the instrument when they were less than 15% of the total dataset [31]. The OriginPro 2020b statistical software was used to perform comparison on the various datasets of interest using the "Compare Linear Fit Parameters and Datasets" application. The application compares datasets by comparing two models: a more complicated model where the parameter value can vary among different datasets and a simpler model where the parameter values are assumed to be the same for all datasets. In the dataset comparison using a fit model, the more complicated model corresponds to the independent fit for each dataset and the simpler model corresponds to a function with all parameters being shared in all datasets. An F-test and associated p-value determine whether the datasets are statistically different from one another, in this case at the 95% confidence level. The "Compare Linear Fit Parameters and Datasets" application was also used to compare the two pXRF preparation methods. Parametric hypothesis testing via a simple F-test for variance and corresponding two-sample *t*-test for means at the 95% confidence level was also performed on the 30-sample subset of AD/ICP-MS measurements and both individual sets of pXRF measurements. The same statistical tests were performed to analyze the effect of soil moisture content on in situ pXRF measurements.

## 3. Results and Discussion

### 3.1. Comparison of Testing Methodologies

In this study, pXRF measurements were taken in situ and for pelletized samples. Processing and pelletizing prior to pXRF analysis optimizes X-ray scattering and mitigates sample variability, as the sample material is homogenized. However, in the case of in situ pXRF analysis, both of these issues are present; as a means of reducing the sample variability during in situ pXRF analysis, multiple measurements should be taken in various spots of the sample material and averaged to obtain elemental concentrations that are representative of the entire sample. In theory, the more in situ pXRF measurements that are taken, the closer the representative concentration values should be to those obtained from pelletized sample material. S1 in the Supplementary Materials show the raw data for the selected metals (Pb, Cr, Se and As). Figure 1 shows the pXRF data for Pb comparing individual in situ pXRF measurements and the average of the three to those of the 30 pelletized samples.

As seen in Figure 1, average in situ values for Pb are closer to the pellet measurements. The scatter of individual measurements around the 1:1 line increases with increasing Pb concentration, which can be attributed to Pb being concentrated in hotspots, i.e., larger particles that are homogenized by sample preparation. However, the average concentration of three measurements is generally close to the 1:1 line. Therefore, the average of the three in situ measurements for each element is used for the statistical comparison of methods.

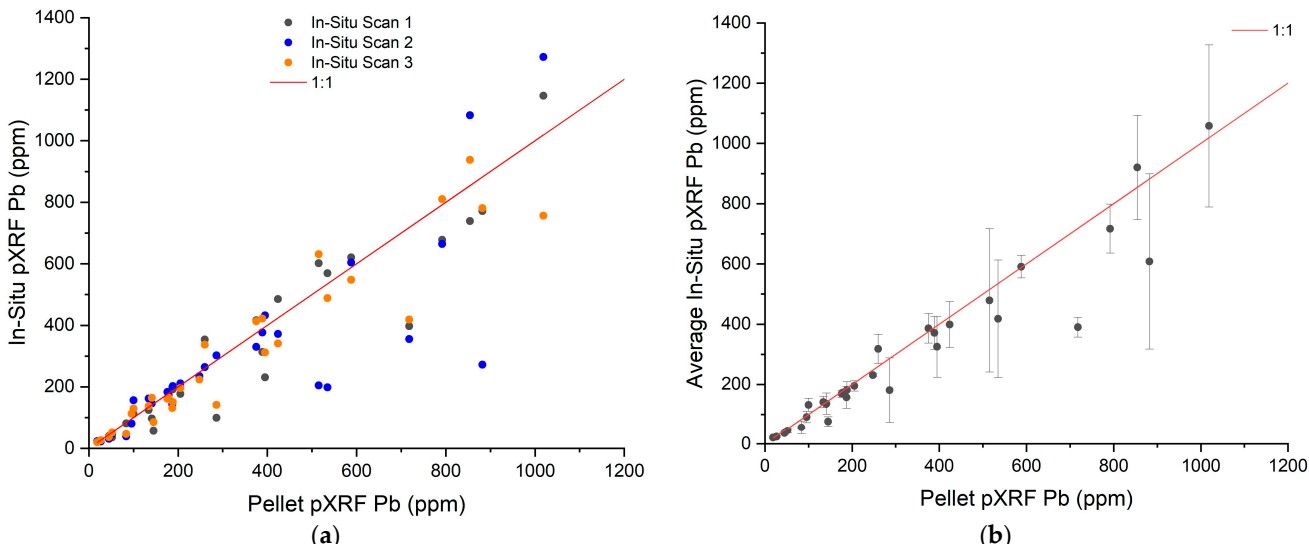

(**a**)  (**b**)

**Figure 1.** (**a**) Individual triplicate in situ pXRF measurements of Pb versus pellet pXRF measurements and (**b**) average of three in situ measurements of Pb with standard deviation error bars versus pellet pXRF measurements, where the red line indicates a regression with a slope of 1.

Figure 2 shows the boxplots for Pb and Cr as determined by in situ and pelletized samples tested by pXRF and AD/ICP-MS, while Table 1 shows the results of statistical tests applied for the two pXRF datasets compared to the AD/ICP-MS one. The two outliers detected in the in situ pXRF dataset for Pb were included in the statistical analyses represented in Table 1, as they were not determined to be outliers in the pellet pXRF or ICP-MS datasets for Pb and the statistical comparison should include all datapoints. How the pXRF compared with AD/ICP-MS depends on the concentration levels, specifically to what extent the concentrations are above the pXRF detection limit. While the equipment manufacturers provide a limit of detection (LOD) for individual elements in a pure silica matrix, these are generally lower than what can be attained in a soil matrix with interferences between elements; this is reflected in the margin of error $\sigma$ reported by the instrument, which is higher than the LOD even when the reported concentration is zero. Thus, the margin of error $\sigma$ is used here in lieu of the theoretical LOD.

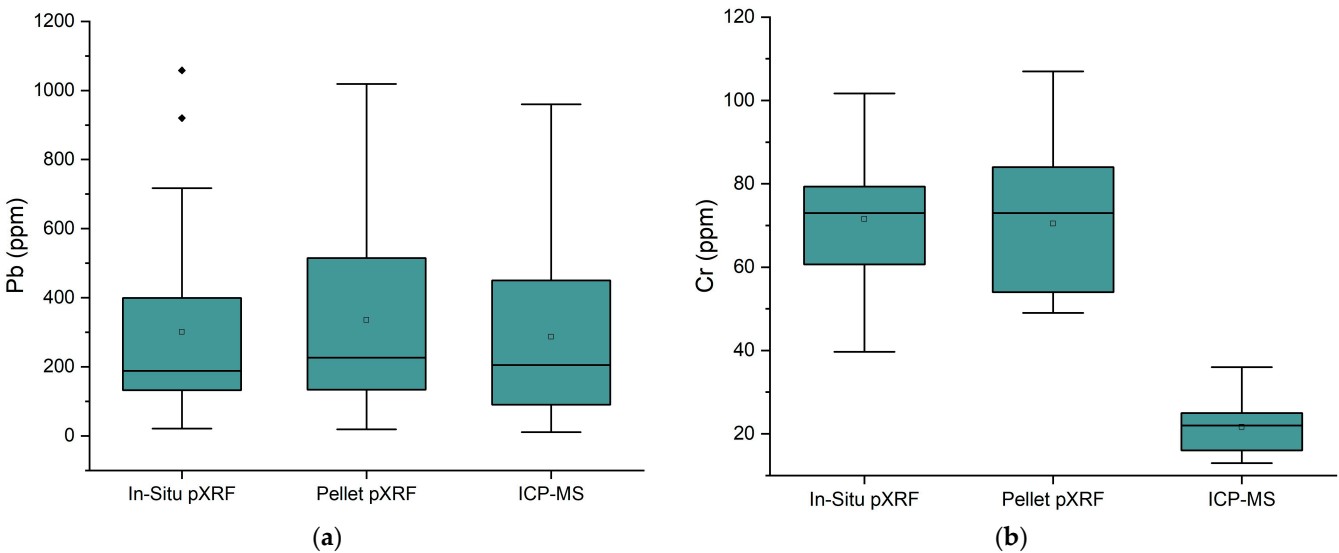

(**a**)  (**b**)

**Figure 2.** Boxplots of metal concentrations as determined by pXRF and AD/ICP-MS for a subset of samples from lot 1: (**a**) Pb; (**b**) Cr.

**Table 1.** Statistical comparison of pXRF and AD/ICP-MS datasets.

| Statistical Test | | Pb | Cr | As |
|---|---|---|---|---|
| In situ pXRF vs. AD/ICP-MS | *t*-test *p*-value | 0.8365 | $1.810 \times 10^{-19}$ | $2.390 \times 10^{-4}$ |
| | F-test *p*-value | 0.6326 | 0 | $4.440 \times 10^{-4}$ |
| | Avg. % difference [1] | 9.941 | 106.3 | 66.16 |
| Pellet pXRF vs. AD/ICP-MS | *t*-test *p*-value | 0.4901 | $1.800 \times 10^{-16}$ | n.a. * |
| | F-test *p*-value | 0.01063 | 0 | n.a. * |
| | Avg. % difference [1] | 21.15 | 104.0 | n.a. * |

* Non-detects account for >15% of the dataset. [1] The percent difference for *x* vs. *y* was calculated by $\frac{x-y}{(x+y)/2}100\%$.

Selenium had concentrations below the pXRF detection limit for all 30 samples. In this case, σ was between 5 and 8 ppm for all samples; AD/ICP-MS confirmed that to be the case for 26 out of 30 samples, with four samples having Se concentrations between 10 and 15-ppm (S1 in Supplementary Materials) that are outliers in the overall distribution. Therefore, the pXRF margin of error is a reliable representation of the actual detection limit and concentration levels of Se in the soil.

Arsenic concentrations were near the pXRF detection limit, with large differences between the two samples preparations: only 2 out of 30 samples were reported as zero in the in situ measurements, whereas 24 out of 30 samples were zero for the pellets; the σ value fluctuated between 2 and 60 ppm (S1 in Supplementary Materials). The AD-ICP-MS measurements yielded concentrations below 8 ppm for all 30 samples; in this case, the instrument's σ value is much larger than the measured values. This is attributed to the overlap between the Pb Lα and As Kα lines, which results in a higher As detection limit for pXRF when substantial Pb is present [32]. Accordingly, the *t*-test and F-tests yielded significant difference between the in situ and AD/ICP-MS dataset, with an average difference of 66%.

For Cr, the pXRF-reported concentrations were five to seven times higher than the σ value for both in situ and pellet datasets. The AD/ICP-MS data indicate that the actual concentrations were much lower for all 30 samples (S1 in Supplementary Materials and Figure 2b); similar to As, the difference is likely attributed to the interference of high Fe content, which is common in soils, with the Cr measurement [32]. Similar to As, the *t*-test and F-tests yielded significant differences between both pXRF datasets and the AD/ICP-MS one, with differences exceeding 100%.

Finally, Pb concentrations are two orders of magnitude higher than the pXRF detection limit, and in this case, the distribution and median values for all three methods are similar, as also corroborated by the *t*-tests (Figure 2a). On average, both methods yielded higher Pb concentrations; as shown in Figure S1, the mean relative difference exceeded zero when compared to the AD/ICP-MS values. However, the scatter for the in situ pXRF method was higher and spanned a larger range, with relative differences between −50% and +60% compared to the AD/ICP-MS (Figure 3 and Figure S1). For the pellet samples, the relative differences ranged from −20% to +65% (Figure S1). In other words, the larger variability of the in situ pXRF measurements caused the F-test to yield a higher p-value when compared to AD/ICP-MS. Higher pXRF measurements compared to AD/ICP-MS imply that the method yields more conservative estimates for the majority of samples. For the in situ dataset, 11 out of 30 points were below the 1:1 line and for the pellet dataset, only five points were below the 1:1 line when compared to AD/ICP-MS (Figure 3). The implication of these findings for evaluation of soil health in this site will be discussed in Section 3.4.

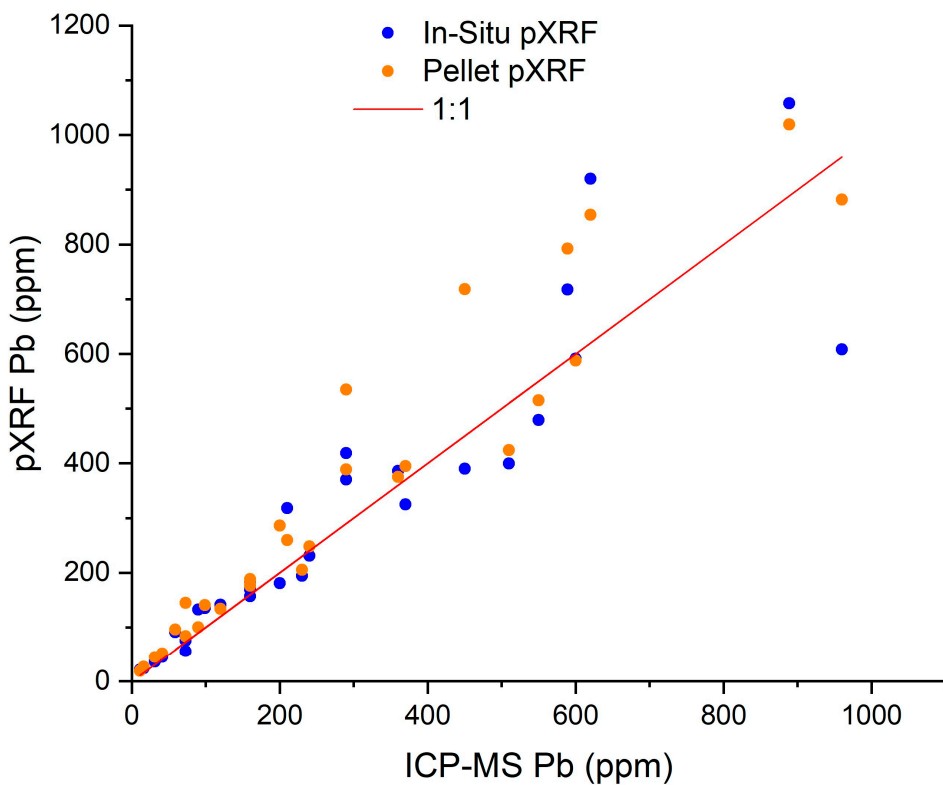

**Figure 3.** Average in situ and pellet pXRF measurements compared to AD/ICP-MS; the red line indicates a regression with a slope of 1.

To further examine the effects of the sample preparation and the difference between the two different pXRF preparation methods, F-tests were conducted for trace metals that were detected above the LOD and are of interest for soil health evaluation, including Ni, Cu and Zn (S1 in Supplementary Materials). Arsenic and chromium were excluded from the analysis, given that they were not found to yield reliable concentrations compared to AD/ICP-MS for this soil. As shown in Table 2, the F-tests indicate that the datasets for Ni, Cu and Zn are different ($p > 0.05$). Specifically, in situ measurements are higher compared to pellet measurements, on average by 30% for Ni, 4.4% for Cu and 8% for Zn. Very few samples showed lower values in the average in situ measurements versus the pellet, and in all cases but one, the difference was less than −10%. Thus, the probability of underestimation of these elements through in situ measurements versus pelletizing is very low.

**Table 2.** Statistical comparison of elemental concentration values determined by in situ and pellet pXRF analysis methods in a subset of samples from lot 1, computed at the 95% CI ($\alpha = 0.05$).

| Element | F-Test for Dataset Comparison *p*-Value | Avg. % Difference [1] |
|---|---|---|
| Pb | 0.08123 | −11.22 |
| Ni | $2.472 \times 10^{-8}$ | 29.47 |
| Cu | 0.01342 | 4.415 |
| Zn | $4.554 \times 10^{-4}$ | 8.116 |

[1] Computed for in situ vs. pellet ($x$ vs. $y$) pXRF measurements as $\frac{x-y}{(x+y)/2}100\%$.

The Pb dataset was different; the *p*-value was higher, indicating less significant differences between the in situ and pellet measurements, and the pellet measurements had several data points higher than in situ (also see Figure 1b). Therefore, for Pb the probability of underestimating concentrations through in situ measurements is slightly higher. The

differences seen between pellet and in situ pXRF measurements can be attributed to the reduction in background radiation and densification of the matrix, which increase the number of scattering atoms per unit volume [33]. The implications of these findings for soil health evaluation in this lot will be further discussed in Section 3.4.

### 3.2. Moisture Content Effect on pXRF Analysis

Moisture content was further investigated as a key variable that can affect pXRF measurements [16,34,35]. Samples from lot 1 contained <5% moisture content, while moisture content in samples from lots 2 and 3 ranged from 7% to 38%. Figure 4 shows the comparison of Pb measurements in wet and dry specimens for 30 samples, with the raw data and associated moisture contents in S2 in Supplementary Materials. Two samples were excluded as outliers, determined as points with residuals greater than 2σ.

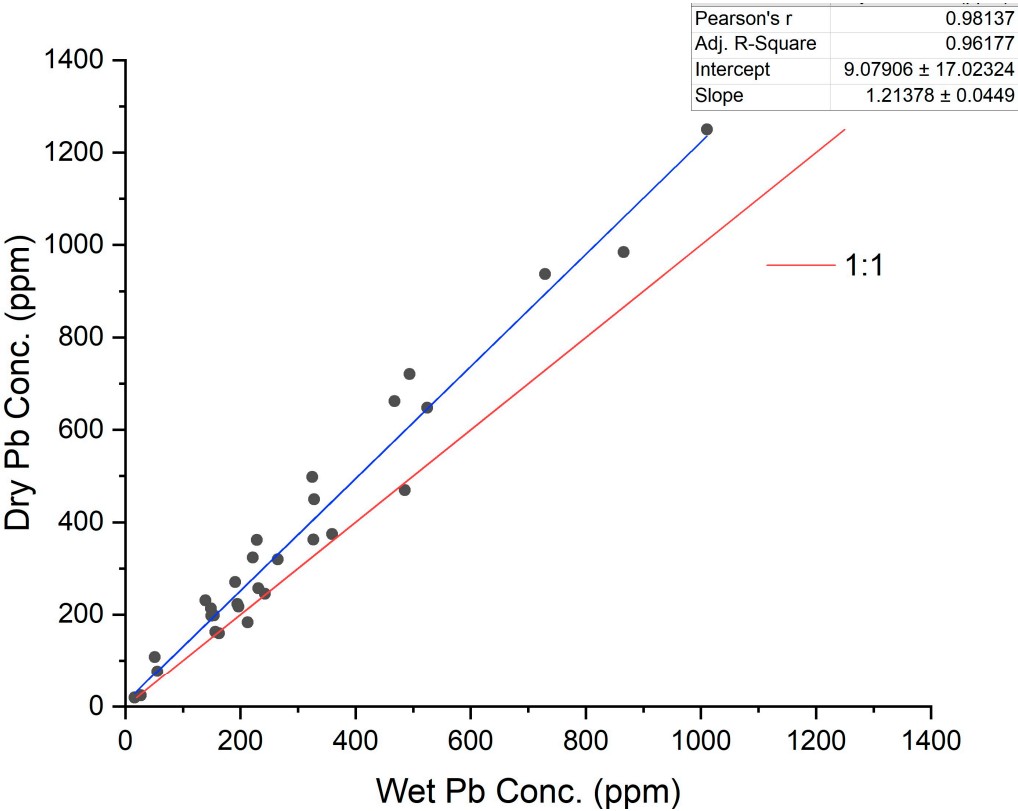

| | |
|---|---|
| Pearson's r | 0.98137 |
| Adj. R-Square | 0.96177 |
| Intercept | 9.07906 ± 17.02324 |
| Slope | 1.21378 ± 0.0449 |

**Figure 4.** Linear correlation between wet (in situ) and dry pXRF Pb measurements (n = 30, $R^2$ = 0.9618) indicated by the blue line; the red line indicates a regression with a slope of 1.

Overall, Pb was lower in the wet samples with the exception of only four samples. As seen in Table S1, the F-tests yielded highly significant differences, with wet samples having on average between 20% and 40% lower concentrations compared to dry samples. Figure 4 represents a linear correlation between wet and dry pXRF measurements of Pb with $R^2$ of 0.96 and a slope of 1.214, which can be used to correct the wet measurements for the dilution introduced by water. As shown in Figure S2, this correction is independent of the measured water content of the samples, which cannot be used to account for the matrix interference introduced by the presence of water. Similar findings can be seen in a study by Foxx [36], who reported a correlation coefficient between wet and dry pXRF measurements of Pb of 0.869 (n = 68) and most points plotting above the 1:1 reference line, with more extreme deviations starting around 250 ppm [36].

To further examine the potential to correct for the effect of moisture content on in situ pXRF measurements, a similar regression to Figure 4 was performed for Ni, Cu, and Zn, since they are trace elements of interest above the pXRF LOD. Two outliers were excluded

from the Cu and Zn datasets, and one for Ni. Figures S3–S5 present the linear regression for each element, plotted against a line with a slope of 1. The regression for Ni has a slope of 1.148 and $R^2$ of 0.87, Cu has a slope of 1.242 and $R^2$ of 0.94, and Zn has a slope of 1.323 and $R^2$ of 0.93. The slopes of the regression between dry and wet measurements for Pb, Ni, Cu and Zn are comparable, confirming that a factor between 20% and 30% can be used to correct for the water matrix effect in these lots.

*3.3. C and N Content*

Total carbon and nitrogen content were analyzed for the purpose of determining soil fertility for growing plants. It should be noted that organic carbon content is often measured for soils and managed by gardeners or famers, while total carbon content encompasses the wide range of carbon species potentially present in soils due to the presence of concrete fragments, black carbon as a combustion byproduct, etc. [37,38]. Likewise, total nitrogen content for soils typically refers to both nitrate and ammonia species of nitrogen. S3 in the Supplementary Materials provides the full dataset of carbon and nitrogen content and the computed carbon-to-nitrogen ratio.

According to the USDA-NRCS, a C:N ratio of 24:1 provides suitable environmental conditions for a soil microbial diet that is most beneficial for plant growth in urban gardens. Out of the 30 samples tested, only three fell within this ideal C:N ratio, with five samples being nitrogen-deficient and 22 samples having a nitrogen surplus or carbon deficit. While nitrogen-deficient soils indicate that plant growth can be notably stunted, soils with excess nitrogen can lead to high weed pressure, excessive plant growth promoting outbreak of harmful insects and mites, excess foliage on fruited plants that reduces yield and delays fruit maturity, and susceptibility to root damage by nematodes and pathogens [39]. A carbon deficiency in soils can also limit crop production by reducing moisture retention, aeration, and key biological functioning of soils [40]. Tactics such as the application of compost can be used to address carbon deficiency in soils. Similar to these results, Lorenz and Lal [41] reported in a review study that total nitrogen concentrations for urban soils are on average 0.21% and 0.08% for 0–10 cm and 20–30 cm depths, respectively. The review also concludes that total carbon concentrations for urban soils are 1.49–3.1% and 0.8% for 0–15 cm and 20–30 cm depths, respectively [41]. Overall, soil amendments are necessary to improve fertility of the analyzed soils for gardening purposes. It should be noted that, should the total carbon and nitrogen concentrations of the soils be higher, additional analyses would be necessary for full screening of soil fertility, including organic carbon, nitrate, and ammonia content.

*3.4. Implications for Soil Health*

The evaluation of different methods has to take into account the context in which they will be used, i.e., the criteria that they will be compared against to determine whether the soil is acceptable for the intended use. There are currently no thresholds established that are specific to soil use for agricultural purposes in the United States or the State of Connecticut. In 2003, the United States Environmental Protection Agency (USEPA) established the Ecological Soil Screening Levels (Eco-SSLs) to report concentrations of common soil contaminants that are protective of ecological receptors, which can be used to identify whether further evaluation is required at the site of interest [42,43]. The USEPA emphasizes that the Eco-SSLs should not be used as cleanup standards, but as a screening-level risk calculation [42]. The Eco-SSLs for plants reported in Table 3 are derived for plants as the ecological receptor of the soil contaminants by considering the exposure pathways of direct contact and plant uptake, and the derived values are stated to exclusively apply to soils with a pH between 4 and 8.5 and organic matter content less than 10%. The USEPA also reports regional screening levels for residential soil. In addition to federal guidelines, soil cleanup standards are developed by state environmental protection agencies, each of which may adopt different models for toxicity and exposure pathways, leading to different

levels. For example, the CT residential direct exposure criterion for lead in soils is 400-ppm, which is also the current USEPA regional screening level threshold for residential soils.

**Table 3.** Federal and CT state soil screening levels.

| Element | EPA Eco-SSL (Plants) [a] | EPA RSL Residential Soil [b] | CT Residential DEC [c] |
|---|---|---|---|
| As | 18 | 35 | 10 |
| Cr (III) | n.a. | 120,000 | 3900 |
| Cr (VI) | n.a. | 230 | 100 |
| Cu | 70 | 3100 | 2500 |
| Pb | 120 | 400 | 400 |
| Ni | 38 | 1500 | 1400 |
| Se | 0.52 | 390 | 340 |
| Zn | 160 | 23,000 | 20,000 |

[a] Ecological Soil Screening Level (Eco-SSL) [43]; [b] Regional Screening Level (RSL) [44]; [c] Direct Exposure Criteria (DEC) [45].

As seen in Figure 5, the evaluation of the three different analysis methods for lot 1 against the regulatory limits for Pb from Table 3 are as follows:

- 400 ppm: Eight samples exceed it for ICP-MS and in situ pXRF versus 9 for pellet pXRF, indicating that pXRF is equivalent or more conservative than ICP-MS at this threshold.
- 120 ppm: 21 samples exceed it for ICP-MS versus 23 for both pellet and in situ pXRF, also showing that pXRF is more conservative than ICP-MS at this threshold.
- The Pb dataset as a whole indicates that pXRF is likely to be more conservative compared to ICP-MS when evaluating soil samples, regardless of the threshold level, which agrees with the statistical analysis that shows pXRF yields higher concentrations compared to ICP-MS (10% higher on average for the in situ pXRF dataset).

This analysis is true when the in situ pXRF measurements are collected on samples that are in a relatively dry state, as were the samples collected for Lot 1. For Lots 2 and 3, the pXRF analysis indicated that the moisture content had a diluting effect on the measured concentrations, with factors ranging from 15% for Ni to 32% for Zn. As shown in Figure 6, this affects the sample evaluation at these sites against the regulatory limits for Pb as follows:

- 400 ppm: Seven samples exceed it in the wet dataset compared to eleven samples in the dry dataset.
- 120-ppm: 28 samples exceed it in both datasets.

Thus, even with the lower concentrations, the probability of missing exceedances in wet samples in these lots is between 17% and 36%, depending on the threshold for Pb in this dataset. The application of the correction factor reduces that probability to 0%, i.e., all exceedances are captured.

For Zn, only the Eco-SSL threshold of 160 ppm is low enough to be comparable with the measured concentrations in the 32 samples; all other values are significantly higher, so that both wet and dry pXRF measurements are well below the threshold. Fifteen samples exceed the Eco-SSL in the dry dataset, and 14 samples in the wet dataset. If the wet dataset is corrected with a factor of 1.3, then 18 samples would exceed the threshold, resulting in an over-correction, i.e., a more conservative evaluation of the dataset.

Similar observations can be made for Cu; 14 samples exceed the Eco-SSL of 70 ppm in the dry dataset, and 11 samples in the wet dataset. A correction with a factor of 1.3 is insufficient to correct for the three samples missed in the wet dataset and results in one other sample to be considered as exceedance; all four samples had concentrations below 100 ppm. All other thresholds are significantly higher than all values in both datasets.

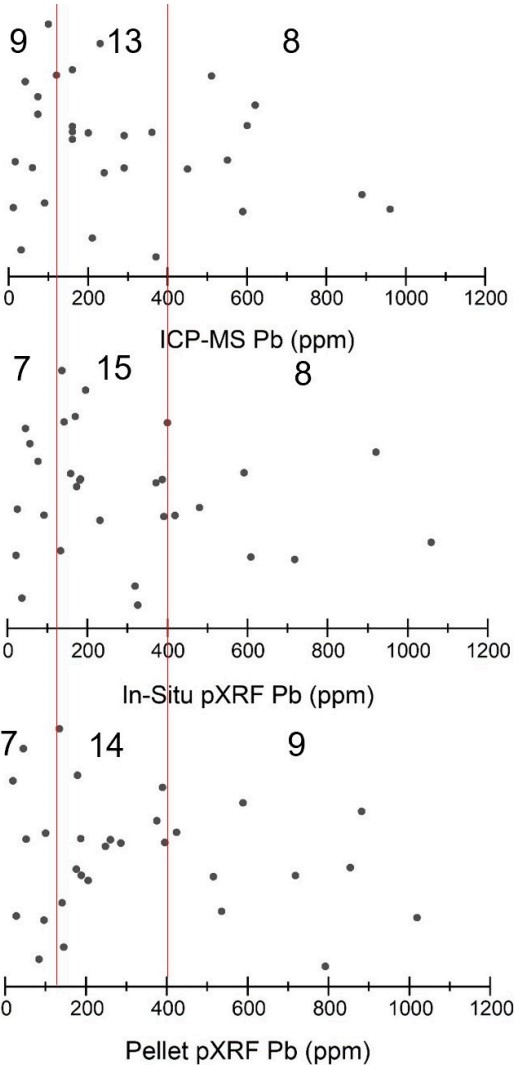

**Figure 5.** Lead concentrations determined by pXRF and AD/ICP-MS analysis for 30 samples with various soil screening thresholds.

For Ni, eight samples exceed the Eco-SSL of 35 ppm in the dry dataset and two samples in the wet; a correction factor of 1.3 results in six samples exceeding the Eco-SSL. Thus, for this low level of concentrations, pXRF is less reliable in terms of accurately capturing exceedances.

Taking all these observations from the study sites into account, it may be said that pXRF correctly captures exceedances in both dry and wet samples when the values are within 30% of the threshold, as long as the threshold exceeds 100 ppm. Therefore, a possible recommendation for in situ field analysis with pXRF in these locations is to analyze samples within 30% of the threshold via AD/ICP-MS. According to this recommendation, eight out of 30 samples would be required to be collected for further laboratory analysis to ensure that Pb levels are below the regulatory limit of 400 ppm. Comparing the Pb levels determined in lot 1 to the USEPA Eco-SSL of 120 ppm for plants, four out of 30 samples would require further laboratory analysis.

In regards to the other selected metals analyzed in this study, the CT direct exposure criteria for soils for Cr, Se, and As are 100, 340 and 10 ppm, respectively. None of the 30 samples show exceedances for these three values; however, there are a few concentration values of As near the threshold. Due to the overlap between the Pb La and As Ka lines, the detection limit of As for pXRF is dramatically increased when substantial Pb is present, which is a commonly abundant trace metal in urban soils; therefore, As cannot be

accurately quantified using pXRF analysis in soils. It should also be noted that due to the overestimation of Cr during pXRF analysis, many of the reported Cr concentrations come very close to the CT DEEP threshold of 100 ppm, with two of the values actually passing the threshold, while AD/ICP-MS analysis only reported Cr concentrations for the entire sample subset from 13 to 32 ppm. Conclusively, pXRF analysis is not a suitable tool for the quantitation of Cr or As at these low levels, and wet chemistry is required. While pXRF analysis of Se yielded mainly non-detect values, the instrumental 1σ error values associated with each non-detect are corroborated by the AD/ICP-MS-reported concentrations, and are well below many of the state and federal screening thresholds, except the Eco-SSL of 0.52 ppm; in that case, AD/ICP-MS must be used to evaluate compliance.

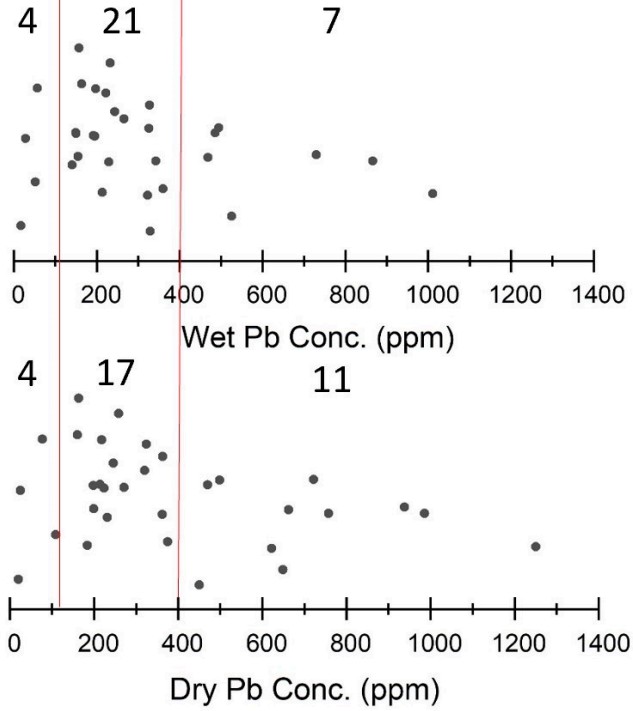

**Figure 6.** Lead concentrations determined by pXRF analysis for 32 wet and dry samples with various soil screening thresholds.

## 4. Conclusions and Recommendations

Portable XRF provides users with the technology to perform a rapid assessment of trace metal content in the field. However, it is imperative to determine the potential interferences introduced by field conditions on XRF measurements and whether field measurements are comparable to traditional laboratory methods prior to the widespread use of portable XRF for soil screening to inform soil management decisions. The evaluation of portable XRF as a method to screen soils for metal contaminants in three test sites in Hartford, CT, with a potential end use as urban gardens yielded the following conclusions:

1. Moisture exceeding 5% introduced a dilution effect that may be accounted for with a correction factor of 1.2–1.3 regardless of the moisture content.
2. pXRF is reliable as a screening tool when used to evaluate exceedances for regulatory thresholds that exceed 100 ppm. For lower thresholds, wet laboratory analyses are recommended.
3. For thresholds exceeding 100 ppm, it is recommended that samples analyzed by pXRF that yield values within 30% of the threshold are also analyzed by wet chemistry.

Further research is needed to determine whether the recommendations for trace metal analysis of soils tested in this study can be extended to a general rapid soil screening methodology for the purpose of promoting urban agriculture.

**Supplementary Materials:** The following supporting information can be downloaded at: https://www.mdpi.com/article/10.3390/su15107924/s1. S1: Raw Data pXRF and ICP-MS Comparison; S2: Raw Data Moisture Content Analysis; S3: Raw Data Carbon and Nitrogen Content; Table S1: Statistical analysis on moisture content data from lots 2 and 3 determined at the 95% CI. Figure S1: Boxplot representation of the relative differences between various analysis methods of Pb. Figure S2. Linear regression for moisture content and dilution factor for Pb measurements, where dilution factor is simply the quotient of the dry Pb concentration and wet Pb concentration. Figure S3. Linear correlation between wet (in-situ) and dry pXRF Ni measurements ($n = 31$, $R^2 = 0.8663$) indicated by the blue line; the red line indicates a regression with a slope of 1. Figure S4. Linear correlation between wet (in-situ) and dry pXRF Cu measurements ($n = 30$, $R^2 = 0.9425$) indicated by the blue line; the red line indicates a regression with a slope of 1. Figure S5. Linear correlation between wet (in-situ) and dry pXRF Zn measurements ($n = 30$, $R^2 = 0.9289$) indicated by the blue line; the red line indicates a regression with a slope of 1.

**Author Contributions:** Conceptualization, M.C., N.B. and H.C.; methodology, H.C., M.C., N.B. and J.I.; software, H.C.; validation, M.C. and N.B.; formal analysis, H.C.; investigation, H.C.; resources, M.C.; data curation H.C.; writing—original draft preparation, H.C.; writing—review and editing, M.C., N.B. and J.I.; visualization, H.C.; supervision, M.C. and N.B; project administration, M.C and N.B.; funding acquisition, M.C., N.B. and H.C. All authors have read and agreed to the published version of the manuscript.

**Funding:** This research was funded by The University of Connecticut's Center for Environmental Sciences and Engineering (4172480).

**Institutional Review Board Statement:** Not applicable.

**Informed Consent Statement:** Not applicable.

**Data Availability Statement:** The data presented in this study are available in the main text and Supplementary Materials.

**Acknowledgments:** We acknowledge Victoria Duffy for participating in the formal analysis and investigation of this study.

**Conflicts of Interest:** The authors declare no conflict of interest.

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
