# Peer review of "A Soil Screening Study to Evaluate Soil Health for Urban Garden Applications in Hartford, CT"

_sustainability, doi:10.3390/su15107924_

Round 1

Reviewer 1 Report

Editor Sustainability,

Dear Editor,

The manuscript - sustainability-2350000 entitled ‘A Soil Screening methodology for soil health to Inform Sustainable Urban Agriculture’ has been reviewed.

The work focused on developing a methodology to test soil health parameters using in-situ screening methods. The manuscript comes in the domain of the journal, and I appreciate the authors for their research work. However, there are several points, as mentioned below, to be addressed if to consider manuscript suitable for publication.

Abstract needs to be improved, for example until the first 21 lines, only method sand background is developed while less attention is given to the findings of the work. The recommendations are made on merely a few analyses as mentioned in manuscript Why the lead was the main focus when authors themselves mentioned about issues related with the trace metal potential exposure. How come a site comprising 639-m2 in size is representative of the whole scenario and complies with the findings of this work in general perspective. More sites in the same area could also be focused to replicate/strengthen the outcomes. Were the outliers form Figure 2 considered in statistical analyses? Line 271, difference in in-situ and pellet measurements was obvious, authors need to discuss the mechanisms behind such findings. Section 3.3, N contents could be discussed in lien with total N but also available N either ammonium or nitrates. Overall, soil health a broad term, author could be specific in title with focus only on findings of this work. The discussion section needs to be overall rewritten, at times it very descriptive and read lose the concentration, it could be focused on how trace metals may or may not be problematic in studied soil samples along with link with soil health.

Line 89 onwards, ‘there is no guidance that unifies such findings to recommend…’ was it related only to US or across the globe? Abbreviate the terms at first these appear in the text, for example, ICP-MS, XRF etc.  Conclusion section also needs to be thoroughly rewritten, currently it’s the repetition of Results, authors should add only the key outcomes of the work.

Sincerely,

comments added

Reviewer 2 Report

The article is a well-written one and has the potential to get published in this journal after the incorporation of some minor revisions as follows: 

1. The introduction part is elaborative depicting the importance of urban agriculture and the need for urban soil screening. However, the author should highlight a brief description of the probable soil particulars for urban gardens that are different from the general agricultural field soil.

2. Methodology part is not properly described. I always encourage the authors to make sub-sections under this part.

3. The author must mention the geographical coordinates of the study area in the methodology part. How did they store the soil samples or the analysis was done just after sample collection? 

4. Mention details about the Olympus Vanta VMR pXRF, and also mention the detailed procedure through which the soil screening was done using this in a separate sub-chapter.

Round 2

Reviewer 1 Report

Dear Editor,

The revised manuscript has been reviewed. Authors could still consider a few of the previous comments. For example, Abstract needs to be improved in line with comments in first round of revision. Another point ‘How come a site comprising 639-m2 in size is representative of the whole scenario and complies with the findings of this work in general perspective’ needs better justification. What about the outliers’ form Figure 2. The discussion section still needs extensive improvement. For further, please see my comments in first round of review. For Conclusion section authors could see any manuscript to phrase the concluding remarks, it still needs to be thoroughly rewritten, currently it’s the repetition of Results, authors should add only the key outcomes of the work.

Sincerely,

Dear Editor,

The revised manuscript has been reviewed. Authors could still consider a few of the previous comments. For example, Abstract needs to be improved in line with comments in first round of revision. Another point ‘How come a site comprising 639-m2 in size is representative of the whole scenario and complies with the findings of this work in general perspective’ needs better justification. What about the outliers’ form Figure 2. The discussion section still needs extensive improvement. For further, please see my comments in first round of review. For Conclusion section authors could see any manuscript to phrase the concluding remarks, it still needs to be thoroughly rewritten, currently it’s the repetition of Results, authors should add only the key outcomes of the work.

Sincerely,
